# Green Processing, Germinating and Wet Milling Brown Rice (*Oryza sativa*) for Beverages: Physicochemical Effects

**DOI:** 10.3390/foods9081016

**Published:** 2020-07-29

**Authors:** John C. Beaulieu, Shawndrika S. Reed, Javier M. Obando-Ulloa, Stephen M. Boue, Marsha R. Cole

**Affiliations:** 1United States Department of Agriculture, Agricultural Research Service, Southern Regional Research Center, 1100 Robert E. Lee Blvd., New Orleans, LA 70124, USA; shawndrika.reed@usda.gov (S.S.R.); steve.boue@usda.gov (S.M.B.); 2Doctorate Program in Natural Science for Development (DOCINADE) and Agronomy Engineering School, Costa Rica Institute of Technology (ITCR), San Carlos Technology Local Campus, PO Box 223-21001, Ciudad Quesada, San Carlos, Alajuela 30101, Costa Rica; jaobando@itcr.ac.cr; 3Department of Chemistry, College of Engineering and Science, Louisiana Tech University, Carson-Taylor Hall, 343, PO Box 10348, Ruston, LA 71272, USA; cole@latech.edu

**Keywords:** green processing, γ-aminobutyric acid, inorganic arsenic, particle size, phenolics, sprouting, viscosity

## Abstract

Plant-based beverage consumption is increasing markedly. Value-added dehulled rice (*Oryza sativa*) germination was investigated to improve beverage qualities. Germinating brown rice has been shown to increase health-promoting compounds. Utilizing green processing, wholesome constituents, including bran, vitamins, minerals, oils, fiber and proteins should should convey forward into germinated brown rice beverages. Rapid visco-analyzer (RVA) data and trends established that brown rice, preheated brown rice and germinated brown rice had higher pasting temperatures than white rice. As pasting temperature in similar samples may be related to gelatinization, RVA helped guide the free-flowing processing protocol using temperatures slightly above those previously reported for Rondo gelatinization. Particle size analysis and viscometric evaluations indicate that the developed sprouted brown rice beverage is on track to have properties close to commercial samples, even though the sprouted brown rice beverage developed has no additives, fortifications, added oils or salts. Phenolics and γ-aminobutyric acid increased slightly in germinated brown rice, however, increases were not maintained throughout most stages of processing. Significantly lower inorganic arsenic levels (113 ng/g) were found in germinated (sprouted) brown rice, compared to Rondo white and brown rice, which is far below the USA threshold level of 200 ng/g.

## 1. Introduction

Sprouted whole grain products have increased markedly in the food and beverage marketplace and re-invigorated health trends [1]. Rice and rice-derived beverages offer a non-soy, lactose-free, cholesterol and gluten free value-added food source. Most rice nutrients are concentrated in the bran fraction, including oils with essential fatty acids, proteins, fibers, vitamins, antioxidants and key micronutrients [2]. Rice bran contains high amounts of fiber and bioactive phytochemicals, such as tocopherols, tocotrienols, oryzanols, B vitamins, phytosterols (β-sitosterol, campesterol, and stigmasterol), carotenoids, and beneficial phenolics. Most of these phytochemicals are recognized to be bioactive compounds that improve human health. These compound roles include antioxidant activities, scavenging free radicals, boosting immune systems, and decreasing the risk of developing heart disease and cancer [3,4,5,6]. Germination and sprouting is an ancient low-cost technology, which increases health-promoting properties [7,8]. Germinating brown rice has been documented to increase the content of γ-aminobutyric acid (GABA) and several antioxidants like γ-oryzanol, vitamin E, phenolic compounds and additional bioactives [9,10,11,12,13,14].

When whole grain rice is eaten, bound and/or undigested phytochemicals are freed by digestive enzymes and micro-flora in the colon to release phenolic compounds that provide health-protective effects in sito and in other locations after transport and absorption [15,16]. GABA has numerous health-beneficial properties in the mammalian diet. These positive effects, studied some time ago, include regulation of heart rate and blood pressure, stimulating immune cells in mice, alleviation of pain, anxiety and sleeplessness in mice, inhibition of cancer cell proliferation and may prevent diabetes in rats, mice and rabbits [17,18,19]. Whole grain cereal phenolics have strong antioxidant activities, which reduce oxidative stresses by scavenging free radicals that potentially damage important large molecules, such as proteins, lipids and DNA [3,15]. Health benefits have been demonstrated with rice bran in animal feeding studies in which diets supplemented with cell wall-bound phenolic fractions resulted in a reduction in hypertension, hyperglycemia, and hyperlipidemia [20], and regulation of blood glucose levels via decreased low-density lipoprotein cholesterol and plasm total cholesterol, signifying effects against type 2 diabetes [21]. Furthermore, simple phenols (i.e., ferulic acid, caffeic acid, etc.) from brown rice extracts had potential chemopreventive action, inhibiting the growth of human breast and colon cancer cells [22].

In recent years, the number of commercially available rice beverages has dropped substantially compared to 5–7 years ago, as alternate plant sources, and “sprouted” seed beverages diversified. However, only rice and soy offer agronomic crop acreages and yields [23], and reduced price-point inputs compared to most other plant-based beverage alternatives (e.g., almond, cashew, chia, coconut, hemp, oat, quinoa, etc.). Interestingly, USA per capita dairy milk consumption has declined steadily from 1975 to 2018 by 40.9% [24]. Plant-based drinks are rapidly becoming a strong segment in the functional beverages marketplace [25]. Global plant-based beverages that are replacing dairy alternatives are expected to surpass USD 34 billion by 2024 [26]. According the Information Resources Inc, plant-based beverage sales increased 6.2% in a year through 2019, and attained USD 1.7 billion sales, yet almond (*Prunus dulcis*) and soy (*Glycine max*) milks command 70.1% and 10.6% of USA market shares, respectively [27].

Several methods and patents exist for making rice beverages and syrup from several starting materials, as outlined previously [28]. However, many of these methods relied upon stabilization of the rice through direct chemical processes (e.g., solvents, heat) or indirectly as byproducts (e.g., bran, press cake, milled portions, flour). Such materials have routinely been used as the process inputs, most often facilitated by enzymatic conversion and/or isolation processes for protein or starch/sugar. Aside from one important patent [29] and revision thereof [30], literature indicating raw materials that were not pre-processed, stabilized or defatted prior to soaking, versus naturally sprouting brown rice to generate beverages through “green processing” was not located.

Consumers are shifting toward clean product labels, with concerns about saturated fat levels, antibiotic and hormone residuals in dairy products, as well as environmental and animal husbandry issues [31]. In response, less processed, healthier, “greener” plant-based products that avoid abusive extraction methods, using proven enzymatic extraction techniques are expanding. A green processing method using germinated brown rice that maintains a liquid, “free-flowing” soluble matrix was developed and defined [28]. With scale-up and pasteurization, the goal is a truly functional natural brown rice beverage with no additives, aside from food-grade enzymes. Herein, physicochemical parameters revolving around starch conversion and rheological properties in germinated brown rice (GBR), through saccharification, delivering initial rice beverages are outlined. Using methods developed specifically for GBR beverages, non-germinated white rice (WR) and brown Rondo rice (BRR) beverages, as well as commercial rice flours (CRF) and beverages (CRB) were compared. Data regarding rapid visco-analyzer (RVA) properties, starch analysis, particle size and viscometric behavior, GABA and phenolic profiles, as well as arsenic concentration changes are presented.

## 2. Materials and Methods

### 2.1. Rice Source, Germination, Thermal Softening, Wet-Milling and Saccharification

Rough and milled Rondo rice were freight shipped to the Southern Regional Research Center (SRRC) and stored at 5 °C until use. Rice growing conditions, harvesting, shipping and moisture analysis evaluations were outlined previously [28]. At the SRRC, rough rice was dehulled (Satake Husk Aspirator, HA 60B, Higashi-Hiroshima, Japan), then manually sorted through three stacked standard sieves (USA #6, #7 and #8, 3.36–2.38 mm, Gilson Co., Inc., Lewis Center, OH, USA) and hand culled, or rice was sorted and graded using a Clipper 400 Office Tester Cleaner (A. T. Ferrell Company, Bluffton, IN, USA). The clipper was used with an 11R (round) top screen and a 10 × 10 wire (square wire mesh) bottom screen with full ventilation blower, and kernels were passed twice.

Methods used to germinate and optimize a free-flowing processing system for softening, milling, gelatinization and initial enzyme treatments were reported previously [28], and are included online as supplemental items (Appendix A). Methods were optimized specifically for the GBR, and also compared against beverages derived from non-germinated BRR and WR. Briefly, freshly dehulled Rondo BRR was soaked at a rice:water ratio of 1:1 then germinated at 35 °C for a total of 48h, with rinsing every 4h, until radicles had emerged, producing GBR. The GBR was thermally softened at a rice:water ratio of 1:2 at temperatures just below the likely gelatinization temperature of Rondo WR (<70 °C) [32,33]. Once softening was complete, a further dilution enabled wet milling in a 4-L blender (Waring Commercial, CB15V, Torrington, CN, USA) to proceed as a “free-flowing” liquid, avoiding gelatinization. Post wet milled (PWM) samples passed a 30-mesh sieve (0.595 mm or 595 µm, Gilson Co. Inc., Lewis Center, OH, USA). Thereafter, crude beverages were purposely gelatinized (80 °C) to facilitate saccharification enzymes using α-amylase (EC 3.2.1.1) and glucoamylase (EC 3.2.1.3) at 300 µL per 100 g theoretical starch at ~55 °C. Post enzyme (PNZ) beverages passed a 140-mesh sieve (0.105 mm or 105 µm, Gilson Co., Inc. Lewis Center, OH, USA) and loss atop the sieves was analyzed (PWM-L and PNZ-L).

### 2.2. RVA (Rapid Visco-Analyzer) Appraisals

An RVA (Model Super 4, Newport Scientific, Warriewood, New South Wales, Australia), using the 7.10 RVA Rice Method, according to the AACCI Method 61-02.01 [34] was used to determine how the BRR samples responded to various preheated temperatures, and to approximate relative gelatinization temperatures based off the pasting temperature and RVA profiles in similar Rondo samples [35]. Commercial rice flours (CRF) were also assessed to gauge differences between experimental samples regarding starch behavior based on peak viscosity, trough, final viscosity and the setback. WR, BRR and GBR samples were freeze dried (Virtis Genesis, 25ES, Pilot Lyophilizer, SP Industries Company, Gardiner, NY, USA) and milled into a flour using a cyclone sample mill (Model 3010-080P, UDY Corporation, Ft. Collins, CO, USA) with a 1.0 mm screen. Three CRF served as comparisons against the in-house WR and pregelatinized BRR. These consisted of a long grain white flour (CRF1, Rivland RL-100, Riviana Foods Inc., Houston, TX, USA), a pregelatinized white flour (CRF2, Remyflo R500P, Remy/Beneo, Morris Plains, Fairfield, NJ, USA), and a high amylose white flour (CRF3, Remyflo R7 150-T). R7 150-T is high amylose rice flour and RL-100 is long grain indica rice flour, and they were chosen due to their amylose contents being close to Rondo [32]. Precooked R500P rice flour was used to view a fully gelatinized rice flour. Viscosity was recorded in rapid visco units (RVUs) wherein each RVU equals 12 centipoise. Breakdown was calculated as peak viscosity–trough, and total setback was calculated as final viscosity–trough.

### 2.3. Starch Determinations

The USDA Starch Research method, which measures total, soluble, and insoluble starch concentrations in a variety of sugar products and by-products, was used to approximate the starch composition in the rice formulas upon processing [36]. Samples were refrigerated for 24–48 h, prior to analysis. Briefly, 10 mL of a diluted, rice slurry was microwaved for 1 min (100% power, 1000 Watt) and sonicated for 5 min at 60% power as previously reported [36]. The treated sample was filtered through a 0.45 µm Whatman PVDF (polyvinylidene fluoride) syringe filter. Then, 800 µL of the diluted sample was mixed with 200 µL of 0.25 M HCl and 1 mL iodometric reagents (1 mM KIO_3_ and 5 mM KI) before measuring the absorbance at 600 nm on a UV-vis spectrophotometer. Approximate changes in starch concentration and percent soluble/insoluble starch were determined against a corn (*Zea maize*) starch standard curve. Analyses were completed in triplicate, per replicate.

### 2.4. Laser Scattering Particle Size Distribution

A Partica LA-950 (Horiba, Irvine, CA, USA) laser scattering particle size distribution was used to assess differences between rice flours, WR, BRR and GBR processed samples. This equipment has a dynamic particle size range from 0.01–3000 µm and is capable of measuring 30 nanometers to 3 mm simultaneously in the same native liquid sample. The three aforementioned CRF were assessed for comparisons. Four commercial rice beverages (CRB) were also assessed. Ingredient labels indicated no added sugars however, the nutrition labels indicated from <1 to 10 g sugars, and an analyses indicated soluble solids ranged from 6.0–10.8° Brix [28]. The anonymous brands were: CRB#1, a non-fortified brown rice beverage; CRB#2, a sprouted, fortified brown rice beverage; and both CRB#3 and CRB#4 were brown rice beverages that were fortified.

### 2.5. Viscometry

The viscosity of rice beverages processed using Rondo WR, BRR and GBR, along with the aforementioned commercial rice beverages, were determined using a Brookfield viscometer (DV II+, Brookfield Engineering Laboratories, Stoughton, MA, USA). Samples were maintained at 25 °C via a recirculating cooling system, and measured using the SC4-18 probe inside an SC4-13RPY sample chamber with 7 mL at 100 rpm. Commercial samples were run at 200 rpm as the torque fell below threshold (10% torque) and, therefore, output centipoise results were converted using Brookfield values for spring tension, spindle and experimental data to calculate centipoise using the equation: (100/RPM) × TK × SMC × Torque, where TK = viscometer torque constant, SMC is the spindle multiplier constant, and Torque and RPM are output data per run.

### 2.6. Phenolics by Folin–Ciocalteu Method and γ-Aminobutyric Acid (GABA) Analysis

Rondo BRR and GBR were freeze dried into powders and WR was milled using a TekMar A-10 Analytical Mill (Janke & Kunkel IKA, Staufen, Germany) into rice powder and frozen (−20 °C) at the SRRC until testing. The Folin-Ciocalteu method was used to determine the phenolic content in the samples [37]. Standard curves of gallic acid (GAE) ranged from 0–31.25 µg/mL and were prepared by adding 0–500 µL to deionized water which was brought up to a total of 1 mL. Preparation started with 0.1 mL of sample plus methanol or water with a 1:10 dilution. Then, 125 µL of GA per standard sample was removed and 125 µL per diluted sample was added to 625 µL of 0.2 N Folin-Ciocalteu phenol reagent in each tube. Samples were vortexed, and allowed to sit at room temperature for 4 min. Then, 750 µL mixture + 500µL of 7.5% Na_2_CO_3_ was added to the solution. They were incubated for 30 min at 40 °C in hot water bath, then, read at 760 nm (Shimadzu UV-1800 Spectrophotometer, Columbia, MD, USA). For GABA, a 10 mL solution was prepared with 89 mM Tris/Na_2_SO_4_ buffer, using 8.9 mL of buffer, 100 µL of 1M DTT, 10.66 mg of 1.4 mM NADP+, 1 mL of 2 mM α-ketoglutarate and 5.6 mg of GABase at 30 °C [38]. Three experimental units were subdivided into nine sample extracts. Samples were dissolved in H_2_O (deionized), vortexed, allowed to settle, and the supernatant was added to designated blanks, samples and 96-well UV plate cells. Serial dilutions were run with 0–10 mM GABA of stock and water; a 10 mM stock solution (20.6 mg) and 10–20 mL deionized water. Blanks used 100 µL of GABase solution, and 90 µL GABase solution plus 10 µL was used for each sample/standard in each well. Plates were incubated for 1 h at 30 °C. The conversion of NADP+ to NADPH was measured as absorbance (340 nm) on a microplate reader (VersaMax, Molecular Devices Co., Sunnyvale, CA, USA). GABA concentrations in samples were calculated from calibration curves using the standard solutions. GABase was a mixture of γ-aminobutyrate glutamate amino transferase (EC 2.6.1.19) and succinic semialdehyde dehydrogenase (EC 1.2.1.16) from *Pseudomonas fluoresces*, purchased from Sigma (St. Louis, MO, USA). Microtiter 96-well UV plates were obtained from Corning (Corning, New York, NY, USA).

### 2.7. Arsenic Evaluations

Arsenic was measured based on the AOAC Method 2013.06, Arsenic, Cadmium, Mercury, and Lead in Foods [39,40], and the updated FDA Elemental Analysis Manual (EAM) (Section 4.11) [41]. Briefly, total arsenic samples were microwave digested with heating blocks using concentrated nitric acid and 30% hydrogen peroxide, prior to separation by an anion exchange and quantification by inductively coupled plasma mass spectrometry (ICP-MS), monitoring the As 75 ion. The quantitation of total arsenic was performed independently to verify the mass balance of the speciation results. A certified reference material (NMIJ CRM 7503a, Arsenic compounds and trace elements in white rice flour, National Metrology Institute of Japan) was included in every test batch to monitor quality control. For arsenic speciation, samples were mixed with a 0.28 M HNO_3_ solution and heated at 95 °C for 90 min, diluted, centrifuged, filtered and analyzed by HPLC-ICP-MS. Arsenic species were separated using an isocratic anion-exchange with an arsenic-specific detector monitoring (*m*/*z* 75), per [41].

### 2.8. Statistical Analysis

Data were analyzed in JMP^®^ 13 PRO for Windows (SAS Institute Inc., Cary, NC, USA) and their distributions were verified by the software so that data points falling outside a normal distribution could be removed. Briefly, the data distribution histograms were viewed. Under the “Percent” option, “Normal Quantile Plot” (JMPs terminology for the Normal Probability Plot) was selected with the “Continuous Fit” designated to “Normal”. The “Fitted Normal” was subjected to a “Goodness of Fit” utilizing the Shapiro-Wilk W hypothesis test to identify data outside the normal distribution, which were manually removed. Thereafter, data were submitted to ANOVA in JMP® 13 PRO for Windows. When statistically significant differences were found, means were compared against the control by the Dunnett’s test at *p* < 0.05. In the cases in which a control was not available, treatment differences were evaluated by Tukey-Kramer HSD (Honestly Significant Difference) test at *p* < 0.05.

## 3. Results

Physicochemical changes resulting from sprouting and creating a brown rice beverage using green technologies and a free-flowing processing are detailed herein. Sprouting was terminated when coleoptile length averaged 2.24 ± 0.83 mm (*n* = 160), which corresponded with a high germination rate of 96.7 ± 0.8% (*n* = 600), and a balance between catabolic loss of macronutrients versus other endogenous physiological changes that might be beneficial, processing-wise and/or health-wise [28].

### 3.1. RVA Characterization of White, Brown and Germinated Rondo Rice and Commercial Samples

Pasting temperature has been associated closely to gelatinization temperature, yet RVA pasting temperatures generally overestimate the rice gelatinization temperature [42]. During processing method development, differential scanning calorimetry was not used to get the exact gelatinization temperature. Instead, we varied the heating temperatures and used RVA to rapidly monitor the extent of pasting in similar experimental samples of WR, BRR and GBR, as compared to heated and non-heated samples (Table 1). RVA pasting profiles can be utilized with care to approximate relative gelatinization temperatures in similar samples [35].

In-house-produced white rice flour had the significantly highest peak viscosities, and final viscosities compared to other WR samples, and likewise, compared to other samples turned into both BRR and GBR (Table 1). In-house WR was of significantly higher or similar pasting temperature compared to commercial WR. The pasting temperature corresponds with the inflection point at the highest viscosity achieved during heating at 95 °C. Subsequently, BRR, and preheated or cooked samples generally, with an offset inflection, had higher or lower pasting temperatures, respectively [28]. For example, during method development, gelatinization was purposely avoided and most BRR samples evaluated with RVA displayed elevated pasting temperature compared to WR. Likewise, preheated BRR at or above the posted temperature range for high amylose long-grain rice and Rondo of 70 °C [32,43], had significantly higher pasting temperatures. Lower peak viscosity in BRR and pre-heated BRR compared to WR samples (Table 1) also indicates that starch granules and proteins began to absorb water, which delayed starch swelling as the temperature increased [44]. During RVA breakdown, the gelatinized paste displays thixotropic behavior that is due to starch molecule alignment or mechanical breakdown of the polymers [45]. Cohesiveness of mass is a cooked rice sensory term that indicates the maximum degree to which a sample rice holds together during chewing, which has a weak correlation with breakdown [46].

BRR and GBR with weaker starch effects and increased chemical interactions from protein, oil and fiber apparently had less swelling power and cohesiveness, as illustrated by significantly lower breakdown. Likewise, it appears that increased moisture and germination effects seem to render a slight swelling and breakdown, even though a completely atypical RVA data profile was observed (Table 1).

Aside from pasting temperature, preheated, cooked and GBR samples had most of the significantly lowest values measured across all parameters, as the WR and BRR tended to deliver more or less normal RVA profiles (Table 1), as illustrated previously [28]. Low final viscosity (694 cP) in cooked commercial WR (Remyflo R500P) and lack of setback indicates that it lacked the ability to gelatinize again (Table 1). This WR exhibited higher initial starting viscosity due to irreversible swelling of starch granules, which, reflects the degree of pregelatinization [47]. On the other hand, the GBR samples with exceedingly low and significantly different RVA parameters, in combination with a significantly higher pasting temperature seems to reflect an interaction due to endogenous enzymatic changes whereby starch swelling and catabolic conversions occurred, in addition to likely chemical interactions with high levels of oil, protein and fiber [28].

Nonetheless, due to several processing times and different experimental trials, not all groupings of Rondo were run, or could be compared, simultaneously. For example, sometimes the brewers WR was used, or BRR would be made and either milled into WR or sprouted into GBR (Table 1). Within subtreatments and occasionally between treatments, there were minor inconsistencies in statistically significance differences between low or high values in given parameters (Table 1). We attribute this to samples having been dehulled on different days, and the possibility that the degree of hull/bran removal may have been slightly inconsistent. De-hulling was accomplished “as gently as possible” in order to facilitate an extremely high germination rate. Subsequently, BRR produced in-house had slight variation due to the nature of the Satake dehusker that forcibly rolls kernels through rollers to remove the hulls, and rice was passed through the Satake twice.

BRR had greater crude protein (7.41 ± 0.25%) and fat (3.59 ± 0.50%) compared to Rondo WR with 6.60 ± 0.04% protein and 1.19 ± 0.06% fat [28]. RVA profile curves, and peak and final viscosities have been demonstrated to be lower in split bean flour compared with a pure white rice flour [47]. This has been attributed to presence of less starch accompanied by greater levels of protein, oil and bran in the bean (*Phaseolus vulgaris*) flour preparation, which would be much lower or exceedingly low in commercially defatted white flour. Subsequently, the lower viscosities reported in BRR samples, compared to WR, in combination with the higher pasting temperatures, indicate that chemical and physical interactions between protein, oil and fiber changed the dynamics of these, and pre-heated BRR’s. The significantly highest pasting temperatures observed (93.8–94.7 °C) occurred in the pre-heated BRR samples (Table 1). Differential scanning calorimetry (DSC) revealed that partial removal of the bran layer via milling to various degrees in brown long grain Lemont rice was found to result in a sharp decrease in gelatinization onset (To), peak (Tp), and conclusion (Tc) temperatures [48]. It was also demonstrated that protein and oil removal from white rice flour with 18 or 24% amylose reduced markedly both peak viscosity and final viscosity [45]. Herein, the RVA data and inferences indicate that the pre-heated (75 and 85 °C) BRR had yet to fully gelatinize, had much higher pasting temperatures, and hence gelatinization temperatures were greater than previously benchmarked (70 °C) in high amylose (26.4–32.2%) white Rondo rice [32,43]. Even though the Rondo WR gelatinization temperature is supposed to be approximately 70 °C, the BRR RVA appraisal indicated that temperatures used in the BRR and GBR experimental processing (~75 °C) did not allow gelatinization during softening and wet milling for beverage processing [28]. The RVA BRR pasting information along with several empirical GBR trials helped establish a free-flowing processing regime (maintained ~75 °C in samples reported herein), without gelatinization, prior to purposeful gelatinization, followed by exogenous enzymatic hydrolysis.

### 3.2. Starch Characteristics

In BRR that was not germinated and subjected to the developed green processing protocol, the PWM samples contained 60.0 ± 2.3 g/100 mL total starch, of which, 97.8 ± 1.0% and 2.2 ± 1.0% were insoluble and soluble starch, respectively. After thermal softening, total starch was somewhat reduced, yet approached the initial level found in BRR (74.3 ± 1.1 g/100g), according to proximate analysis [28]. After gelatinization, hydrolysis with two amylolytic enzymes and passing through a 140-mesh sieve, total starch levels markedly declined. The BRR PNZ rice beverages contained only 22.4 ± 6.7 g/100 mL total starch, of which, insoluble and soluble starch percentages (97.6 ± 1.5% and 2.4 ± 1.5%, respectively) were not changed appreciably. The mass balances from starting starch through the PNZ were subsequently off. There was minor processing loss atop the 140-mesh screen and the majority of this material was likely larger grain starch, and fiber (based on color properties [28]). Moreover, in enzyme-treated beverages passing the 140-mesh, we believe some oligosaccharides were still in solution and hydrolyzed into smaller soluble chains that were not picked up by the starch iodine staining in the method [36]. This method has optimal kinetics validated for low concentration starch impurities in sugarcane and, because it is based on corn starch, it may not translate well into GBR rice starch where endogenous oligosaccharides, fiber, protein and oil are abundant. Since both the BRR and especially WR “control” samples processed with the optimized free-flowing green process had significantly greater starting starch concentrations [28], the soluble/insoluble starch analyses were not always reliable, and not analyzed across all treatments.

### 3.3. Particle Size and Viscometry

GBR samples that were gelatinized, enzyme treated and passed a 140-mesh sieve (PNZ) generally had significantly larger particle size parameters than the post-wet milled (PWM) crude beverages that passed 30-mesh sieve prior to gelatinization (Table 2). Not including commercial samples (CRB), the lowest D10 cumulative diameter (5.7 µm) and median size particles (10.0 µm) were observed in GBR PWM samples (Table 2). The PWM samples passed a 30-mesh sieve (595 µm), whereas the PNZ samples passed a tighter 140-mesh sieve (105 µm). Mean particle size, D50 median particle size (splits the diameter distribution) and D90 (cumulative percentage of 90% diameters) were significantly higher in GBR PNZ compared to PWM. Aside from D10, the aforementioned means were significantly higher in GBR PNZ samples than both WR and BRR PNZ samples (Table 2). This indicates that only gelatinized GBR PNZ rice samples delivered through endogenous and exogenous enzyme activities experienced significant size increases. Meanwhile, all three beverages produced from WR, BRR and GBR contained roughly ~15% soluble solids [28], with oils and protein abundantly present in BRR and GBR (Beaulieu, unpublished data [49,50]). Subsequently, in GBR PNZ, there could be copious water hydration facilitating electrostatic interactions and/or hydrogen bonding or noncovalent complexing (e.g., ionic surfactant or dispersion interaction) between oligosaccharides, oil, fiber, sugars and protein, resulting in larger particles compared to other experimental samples. Apparently, chemical interactions between protein, oil, fiber and soluble solids along with smaller oligosaccharides continued to occur only in the GBR PNZ system, which led to an apparent unstable emulsion. This runs contrary to normal emulsification principles (Stoke’s law) whereby smaller particle sizes are known to aid in stability of oil–water or water–oil emulsions [51]. However, this too is contrary to previous findings indicating that use of stabilized rice bran or flour with exogenous enzyme hydrolysis leads to isolated rice protein with poor solubility in water and relatively poor emulsification properties [52,53]. We presume the endogenous cascade of enzymatic events in GBR and altered RVA profile may indicate physicochemical interactions and behavior that are generally not observed in similar commercial beverages, which have not included truly germinated materials. This supposition will be further explored elsewhere (Beaulieu, unpublished data [49,50]).

In general, the two native white rice flours had larger particle sizes than in-house WR samples. Yet, a fully gelatinized (cooked) white flour (Remy R500P) had the highest particle sizes compared against all other tabled means (Table 2). The two commercial WR flours tested had bimodal particle size distribution peaks at 10 µm and 100 µm (Figure 1; e.g., R7 150-T), whereas the pregelatinized rice flour (Remy R500P) had generally a single peak distribution closer to 500 µm, which was also wider (Figure 1). Particle distribution profiles also illustrate markedly increased particle sizes observed in only the post enzyme-treated (PNZ) GBR samples, compared to the original PWM GBR (Figure 1). This again seems to indicate chemical interactions between natively catabolized complex classes of compounds that have been carried forward into a natural beverage, after gelatinization and exogenous enzyme liquefaction.

Viscosity data indicate that both WR and BRR PWM samples were significantly more viscous than GBR PWM samples (Table 2). Meanwhile, both GBR and BRR PNZ viscosities were significantly lower than the WR PNZ samples, which had the highest viscosities in the developed beverages (Table 2). Interestingly, the GBR PNZ samples displayed much larger particle sizes, yet they were significantly less viscous than BRR PNZ. In theory, a white rice flour or WR-derived beverage would be the product of almost exclusively starch hydrolysis, containing little to no natural fat, protein and fiber compared to BRR that could contribute to chemical interactions which, as previously mentioned, have greater probability to occur in GBR samples. This is deduced based on the fact that both BRR and GBR PNZ beverages had significantly lower calculated Brookfield viscosity, while simultaneously, only the GBR PNZ beverages had a significantly higher mean, D50 and D90 particle sizes (Table 2 and Table 3). It will be useful to collect additional shear rate ramp and rpm speed rate ramp viscometer data during scale-up in the pilot plant as processing parameters are brought to completion (e.g., after emulsification and/or homogenization followed by pasteurization).

In general, aside from the viscosity measurement of the GBR CRBs, three of the four commercial rice beverages analyzed had very similar particle size parameters (Table 2). This is not surprising since most commercial products have been industrially streamlined for homogeneity, with added oils, salt, fortification, and have been homogenized and pasteurized. CRB#1 was a non-fortified brown rice beverage, but with the exception of D50, all of its particle size and viscometric properties were extremely similar to the other two commercial brown rice beverages (CRB#3 and CRB#4), which were fortified. CRB#2 was a “sprouted”, fortified brown rice beverage and it displayed similar particle size characteristics as the other three commercial beverages, and only significantly different (larger) D10. 

However, CRB#2 had significantly higher viscometric measures compared to the other three commercial samples (Table 2). According to the ingredient labels, no gums like gellan or gum Arabic were added to these commercial samples. Although speculative because commercial processes and level of soaking versus sprouting is unknown, it seems interesting how both the developed (GBR, PNZ) and commercial sprouted beverage (although fortified) had similar viscometric properties. The experimental WR crude (PWM) and post-enzyme treated (PNZ)-samples had D90 cumulative particle sizes that were very close to both BRR PNZ samples and CRB samples (Table 2 and Table 3). It is not fully understood why the BRR PWM samples contained the significantly greatest D90 particle size of the study (aside from the fully gelatinized commercial white flour). However, it should be noted that the CRB samples all had exogenous oil added, and three contained additives (fortification). Hence, comparisons concerning moiety interactions and particle sizes in CRBs are speculative. Nonetheless, the in-house enzyme-treated GBR beverages had somewhat similar viscometric means compared to those commercial samples. These relationships indicate that soaking and germination-related physiological and/or chemical changes to protein, oil and fiber in the matrix (e.g., delivering soluble solids and oligosaccharides though both endogenous and exogenous catabolic enzyme activity) resulted in the developed natural GBR beverages to deliver rheometric properties similar to a commercial “sprouted” brown rice beverage. During informal sensory appraisal of the CRBs, the sprouted brown rice beverage was found to have the highest “viscosity”, but also had the highest “chalkiness”, “starchy”, “nutty” and “grassy” attributes (Appendix A). Likewise, the sprouted CBR was also found to have the lowest “pleasant”, “sweetness”, “mouth coating” and “creamy” attributes. None of the CRBs had sugar added. Unfortunately, in-house beverages were not evaluated because they were not pasteurized.

Due to apparent interaction effects observed (e.g., Table 2), rice type and treatment effects were evaluated and presented as a factorial analysis for the D90, mean particle size and calculated viscosity. Data indicate that there were significant main effects based on rice types (GBR, BRR and WR), and interaction effects based on processing (PWM and PNZ) in D90 (Table 3). In general, GBR and BRR rice types behaved similarly and were significantly different from WR, and this was expected. The interaction effects indicate that GBR particle size and viscosity in PNZ samples were often significantly different from WR, and generally significantly different from BRR. For example, the effect of whether or not the enzyme digestion (PNZ) will be different compared to the controls (PWM) was highly dependent upon the rice type; especially WR as opposed to both BRR and GBR, and again especially dependent upon only GBR (Table 3), since these were the only germinated samples.

### 3.4. γ-Aminobutyric Acid (GABA) and Phenolics

GABA increased in all experiments when BRR was germinated into GBR, whereas phenolics increased in three of four experiments (Table 4 and Appendix A). For example, there was an 11.0–26.0% increases in GABA. The GABA concentrations recovered (0.222–0.868 mg/g; raw data range) fell within ranges previously reported in rice [9,10,11,54,55]. Phenolics recovered from BRR and GBR samples throughout the processing ranged from 1.18–11.90 (mg/g; raw data range). These levels fell within ranges comparably to those in the literature regarding rice [9,56,57]. Increased phenolics were generally observed when BRR was germinated into GBR (between 4.3–22.9%). However, the results for phenolics in one trial were opposite to the normal reported trend whereby certain health-beneficial compounds increase due to germination (Appendix A, italicized entry) and, subsequently, the phenolic means tracked through the green processing of BRR into GBR, and onward, were not significant when outliers were processed through ANOVA (Table 4).

The literature indicates clearly that health-beneficial compounds like GABA are enhanced in rice by various germination regimes [8,9,10,11,14,54,55,57,58,59]. However, there is wide variability in reported temperature and duration of germination [8]. Oftentimes, the literature has differing results concerning increased or decreased levels of key nutrients based on grain type, variable soaking/germination time and condition, in addition to post-processing differences [9,10,60,61,62,63,64]. The physiological effects of germination are likely affected by the presence versus absence of the hull. Occasional studies illuminate these experimental conditions (e.g., [10] is paddy rice), whereas it is not clearly denoted in many other works. Furthermore, there may be a variety of effects which have been well studied in Asian varieties [54,63] but not studied adequately in long grain varieties. The maximum germination temperature associated with maximizing GABA concentrations in one short grain japonica and one long-grained indica brown rice was 35 °C, and temperatures above that resulted in GABA decreasing [14]. On the other hand, increased germination temperature (34 versus 28 °C) negatively affected GABA and total phenolic accumulation in two long grain brown rice varieties, whereas similar effects were not observed in two other long grain varieties [9]. Methods used herein purposely pushed the germination temperature to an “upper limit” (35 °C) to speed up the overall process. There is scant information related to tracking some of these nutrients through multi-step processes after germination, such as beverage production that uses markedly higher processing temperatures compared with germination. Thermal processing treatments (softening, wet milling and gelatinization) also decreased slightly (yet insignificantly) GABA and phenolics (Table 4). This was similar to a previous report where parboiling rough and brown rice decreased the concentrations of total phenolics, yet the concentrations of lipophilic antioxidants increased [65]. As this green processing method relies upon wet milling and diluting the samples to maintain a free-flowing state, the concentration of soluble phenolics could be significantly affected by the protocol and heat-effects, as similarly reported [57,65]. However, all three PNZ beverages had significantly increased GABA concentrations after exogenous enzyme treatment (Table 4). Future studies should perform a time course evaluation of several health-beneficial compounds (e.g., phenolics, GABA, ferulic acid, γ-oryzanol) during lower temperature germination and through beverage or value-added treatments to better gauge processing effects. Likewise, future appraisal of developed rice beverages after homogenization and pasteurization will also be necessary to determine the ultimate fate of health-promoting compounds. Loss of health-beneficial compounds like phenolics and anthocyanins is a common occurrence that has been demonstrated in beverage production schemes utilizing enzymes and heat pasteurization [66,67].

### 3.5. Arsenic Evaluations

The Food and Agriculture Organization of the United Nations and Codex Alimentarius Commission of the World Health Organization have set a limit of 200 ng/g (ppb, *w*/*w*) inorganic arsenic for polished (white) rice, and the European Union and USA FDA have proposed stricter limits, where infant rice food levels will be 100 ng/g [68]. The arsenic speciation in several Rondo rice types was therefore evaluated. Only rough rice had inorganic arsenic concentrations above the threshold (Table 5). Importantly, all other inorganic arsenic levels were below the threshold level. Inorganic arsenic levels in the GBR that had experienced rinsing and germination were below the threshold (0.113 mg/kg, or 113 ng/g) and the significantly lowest level found in the study (Table 5). All arsenic levels observed (total, organic and inorganic) in Rondo GBR were significantly lower that all other samples (BRR, WR and rough rice). Chronic exposure to inorganic arsenic is closely associated with negative health effects [69]. Arsenic accumulates in the rice bran and concentrations therein may be an order of magnitude higher than the bulk grain [70]. Therefore, significant reductions in inorganic arsenic in GBR are positive findings concerning their use, especially considering that further dilution also occurred in the free-flowing processes described herein.

## 4. Conclusions

Utilizing the data reported to date, an efficacious and effective free-flowing green process to germinate (truly sprout), soften, wet mill, gelatinize then saccharify brown rice into beverages was developed. The “green”sprouted brown rice beverage developed has no additives, fortifications, or added oils or salts. The method has very low inputs, is rather simple in terms of required equipment, and is applicable for both germinated brown and colored rice varieties. Based on rheological data, observations and preliminary evidence, protein and oil remain in the germinated brown rice beverages, and appear to be held in an unstable emulsion. Protein and oil characteristics in these beverages will be reported elsewhere, assessing levels retained in the developed beverages and the solubility properties. Although phenolic data were occasionally inconclusive, the GABA was found to increase slightly through germination, and significantly increase after saccharification of all in-house beverages. Whether or not health-beneficial compounds can be maintained without significant concentration loss in final beverages remains to be determined. Future analysis of other compounds besides GABA and phenolics should also help answer this question. The GBR post-enzyme-treated beverage had viscosity and particle size characteristics that were similar to commercial rice beverages. Inorganic arsenic was also found to be significantly reduced through germination (0.113 mg/kg), and should not be of concern, even at reduced threshold levels, after dilution and final beverage production. The process will next be scaled-up in the pilot plant to facilitate commercial-like pasteurization.

## Figures and Tables

**Figure 1 foods-09-01016-f001:**
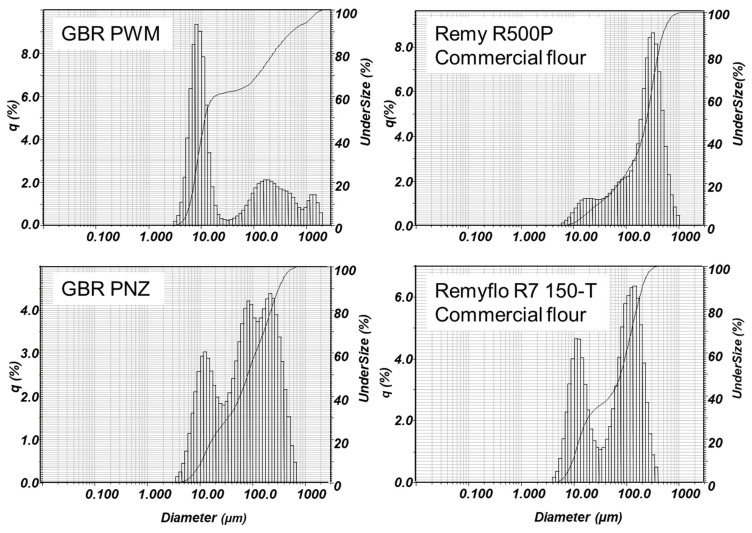
Typical Horiba particle size distribution plots for germinated brown rice (GBR), post wet milling (PWM) and post-enzyme (PNZ), and two commercial white flours: Remyflo R7 150-T and pre-cooked Remy R500P. The y-axis q (%) indicates the amount of each size by volume.

**Table 1 foods-09-01016-t001:** Rapid visco-analyzer (RVA) results illustrating freshly prepared Rondo rice flours: white rice, brown rice and germinated brown rice, and commercial samples with “cooked” and “preheated” temperature effects.

	Peak Viscosity ^†^		Trough		Break Down		Final Viscosity		Setback		Total Setback		Pasting Temp (°C)	
White rice (WR ^‡^)														
WR (Run-1)	3788	a ^§^	3549	a	240	e	7511	a	4088	a	4328	a	90.5	a
WR (Run-2)	3245	a,b	2798	b	447	d	5226	b,c	1981	c	2428	c	87.6	b
WR, CRF-1	2810	b	1751	c	1059	b	4390	c	1580	d	2639	c	87.2	b
WR, CRF-3	3082	b	2395	b	687	c	5480	b	2398	b	3085	b	87.1	b
WR, CRF-2 (cooked)	1758	c	504	d	1254	a	694	d	n.d.		190	d	72.5	c
Brown rice (BRR)														
BRR (Run-1)	2067	a,b	2046	a,b	21	b	4181	a	2114	a	2135	a	90.7	b
BRR (Run-2)	2754	a	2337	a	417	a	4501	a	1747	a,b	2164	a	87.8	c
BRR preheated, 75 °C	2093	a,b	1894	a,b	199	a,b	3171	a	1079	b	1277	a	93.8	a
BRR preheated, 85 °C	2086	b	1724	b	362	a	3221	a	1135	b	1497	a	94.7	a
Polished or sprouted BRR														
BRR → WR (Run-1)	3788	a	3549	a	240	b	7877	a	4088	a	4328	a	90.5	b
BRR (Run-1, control)	2067	b	2046	c	21	c	4181	b,c	2114	b	2135	b	90.7	a,b
BRR (Run-3, control)	1536	b	1543	c	n.d.		3703	c	2167	b	2159	b	92.3	a
BRR → GBR (Run-3)	305	c	135	d	169	b,c	352	d	48	c	217	c	92.2	a,b
Preheated or cooked														
BRR preheated, 75 °C	2093	a	1894	a	199	b	3171	a	1079	a	1277	a	93.8	a
BRR preheated, 85 °C	2086	a	1724	a	362	b	3221	a	1135	a	1497	a	94.7	a
WR, CRF-2 (cooked)	1758	a	504	b	1254	a	694	b	n.d.		190	b	72.5	b

**^†^** All columns aside from pasting temperature are centipoise (cP), as derived from the instrumental output rapid visco units where centipoise = (rapid visco units (RVUs) * 12). **^‡^** Acronyms for treatments are: WR, white rice; CRF, commercial rice flour; BRR, brown Rondo rice; GBR, germinated brown rice. Hyphenated run numbers indicate separate sample preparations from different days of dehulling. **^§^** Means not connected by same letter, within each group of treatments, are statistically significantly different according to Tukey-Kramer HSD test at *p* < 0.05; n.d. indicates not detected (or calculable).

**Table 2 foods-09-01016-t002:** Particle size distribution (Horiba) and viscometric (Brookfield) differences in Rondo rice that was milled or dehulled, germinated, and then moved through a free-flowing green process, resulting in rice beverages.

Rice Type	Treatment	Median Size ^†^ (D50, µm)	Mean Size (µm)	D10, Diameter on Cumulative 10 % (µm)	D90, Diameter on Cumulative 90 % (µm)	Viscosity (cP)	Shear Stress	Calculated Viscosity (cP)
GBR ^‡^	PWM	10.0 b,B,^§^	65.1 b,B	5.7 b,B	235.1 a,B	14.2 a,B	18.8 a,B	1208.7 a,B
	PNZ	78.4 a,Z	120.7 a,Z	10.6 a,Y	296.8 a,Z	2.9 b,Y	3.2 b,Y	243.2 b,Y
BRR	PWM	75.2 a,A	123.1 a,A	8.3 b,A	311.3 a,A	16.4 a,A	21.7 a,A	1405.7 a,A
	PNZ	45.4 b,X	67.0 b,X	11.1 a,Y	159.2 b,X	3.1 b,Y	4.0 b,Y	262.8 b,Y
WR	PWM	84.6 a,A	96.6 a,B	7.8 b,B	189.0 a,B	16.1 b,A	21.0 b,A	1308.7 b,A
	PNZ	66.7 b,Y	86.5 a,Y	13.7 a,Z	191.2 a,Y	19.4 a,Z	25.9 a,Z	1433.0 a,Z
BRR	CRB#1	65.6 a	61.3 a	2.5 b	134.5 a	1.9 b	5.2 b	166.0 b
GBR	CRB#2, fort. ^¶^	47.4 a,b	59.2 a	5.6 a	134.4 a	4.6 a	12.1 a	392.5 a
BRR	CRB#3,#4, fort.	29.0 b	53.0 a	3.0 b	130.3 a	1.9 b	5.0 b	161.3 b
CRF	White flours	74.8 b	88.1 b	8.6 b	192.5 b	---	---	---
	Cooked flour	244.7 a	260.7 a	29.0 a	508.3 a	---	---	---

**^†^** Median size (D50) is an aggregate particle size through 50% of the population; D10 and D90 are 10 and 90% of the cumulative population, respectively; viscosity (cP) is reported in centipoise, SS is shear strength; and calculated viscosity (cP) represents the manufacturer’s equation to calculate and compare viscosities (as stated in the Materials and Methods Section). **^‡^** Acronyms for treatments are: GBR, germinated brown rice; BRR, brown Rondo rice; WR, white rice; CRF, commercial rice flour; PWM, post wet milling; PNZ, post saccharification enzymes; CRB, commercial rice beverages and CRB fort., commercial rice beverages fortified. **^§^** Means not connected by the same lower case letter, per rice type, are significantly different according to a Tukey-Kramer HSD at *p* < 0.05. Means per treatment (processing step) that are not connected by the same upper case letters (beginning of the alphabet for PWM versus ending of the alphabet for PNZ) are significantly different according to Tukey’s test at *p* < 0.05. **^¶^** BRR CRB#1 was not fortified. GBR (“sprouted”) CRB#2 was fortified. BRR CRB#3 and #4 represents two fortified brands combined.

**Table 3 foods-09-01016-t003:** Factorial analysis for rondo rice type and treatment effects in selected particle size and viscometer parameters in germinated brown rice beverages produced through green technologies, compared to brown rice and white rice subjected to the same processing, without germination.

Horiba Particle Size, D90 Diameter on Cumulative % (µm)					Horiba Particle Size, Mean Size (µm)					Brookfield Viscometer, Calculated Centipoise (cP)				
Rice type (Rt)					Rice type (Rt)					Rice type (Rt)				
GBR ^†^	265.9	a			GBR	92.9				GBR	726.0	b		
BRR	235.3	a			BRR	95.0				BRR	834.2	b		
WR	190.1	b			WR	82.7				WR	1406.9	a		
Treatment (T)					Treatment (T)					Treatment (T)				
PWM	245.2	a			PWM	90.7				PWM	1331.7	a		
PNZ	215.8	b			PNZ	87.7				PNZ	646.3	b		
Interaction (Rt×T)					Interaction (Rt×T)					Interaction (Rt×T)				
	PWM		PNZ			PWM		PNZ			PWM		PNZ	
GBR	235.1	b,c	296.8	a,b	GBR	65.1	b	120.7	a	GBR	1208.7	a	243.2	b
BRR	311.3	a	159.2	d	BRR	123.1	a	67.0	b	BRR	1405.7	a	262.8	b
WR	189.0	c,d	191.2	c,d	WR	79.0	b	86.5	c	WR	1380.7	a	1433.0	a
Significance level ^‡^					Significance level					Significance level				
Rt	****				Rt	NS				Rt	****			
T	*				T	NS				T	****			
Rt × T	****				Rt × T	****				Rt × T	****			

**^†^** Acronyms for treatments are: GBR, germinated brown rice; BRR, brown Rondo rice; WR, white rice; PWM, post wet milling and PNZ, post saccharification enzymes. **^‡^** Data were submitted to a factorial analysis with rice type (Rt) and treatment (T) as factors. In the cases in which the interaction was statistically significant (*p* < 0.05; *p* < 0.01; *p* < 0.001; *p* < 0.0001 (*; **; ***; ****; respectively), the mean data were submitted to a Tukey-Kramer HSD test (*p* < 0.05) to identify differences. The means not connected by the same letter are significantly statistically different. In cases where the interaction was not significant (NS), the means of the factors were submitted to a Tukey-Kramer HSD test at *p* < 0.05. Means not connected by the same letter are significantly statistical different.

**Table 4 foods-09-01016-t004:** Statistical analysis of γ-aminobutyric acid and phenolic mean differences assessing the effects of processing brown Rondo rice into beverages.

Rice	Treatment	GABA (mg/g)	Phenolics (GAE, mg/g)
GBR ^†^	BRR (control)	0.41	7.44
	GBR	0.49	7.79 (8.33) ^‡^
	PWM	0.39	7.94 (8.23)
	PWM-L	0.41	7.13
	Gelatinization	0.35	6.51
	PNZ	0.68 * ^§^	d.r. ^¶^
BRR	BRR (control)	0.42	10.48
	PWM	0.28 *	2.43 *
	PNZ	0.67 *	d.r.
WR	WR (control)	0.23	2.97
	PWM	0.63 *	0.02 *
	PNZ	0.59 *	d.r.

**^†^** Acronyms for rice type and treatments are: GBR, germinated brown rice; BRR, brown Rondo rice (non-germinated); WR, white rice; PWM, post wet milling; PWM-L indicates the lost materials on the 30-mesh sieve; gelatinization was a control sample prior to enzyme treatments and PNZ, post saccharification enzymes; GABA, γ-aminobutyric acid. **^‡^** Parenthetical values are placed to illustrate the means if all data were used (see discussion, and S4 Table) to contrast the actual JMP statistical approach where outliers were removed to assume a normal distribution. **^§^** Means highlighted with * are significantly different from the BRR control or WR control according to Dunnett’s test at *p* < 0.05. **^¶^** d.r. = data removed due to presumed colorimetric interference between enzymes and the Folin–Ciocalteu reagent.

**Table 5 foods-09-01016-t005:** Concentration of arsenic species in Rondo rice, as tracked from stored rough rice (paddy) through dehulling, milling and after rinsing and sprouting, resulting in germinated brown rice.

Processing Step/Treatment	Total Arsenic (mg/kg)	Inorganic Arsenic (mg/kg)	Organic Arsenic (mg/kg)
GBR ^†^	0.237 * ^‡^	0.113 *	0.140 *
Controls			
BRR	0.463 a	0.199 b	0.226 a
WR	0.413 a	0.160 b	0.217 a
Rough	0.507 a	0.317 a	0.193 a

**^†^** Acronyms for treatments are: GBR, germinated brown rice; BRR, brown Rondo rice and WR, white rice. **^‡^** Means highlighted with * are significantly different from the BRR (control) according to Dunnett’s test at *p* < 0.05. Means not connected by the same letter are significantly different according to a Tukey-Kramer HSD at *p* < 0.05.

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
