# Peer review of "Green Processing, Germinating and Wet Milling Brown Rice (Oryza sativa) for Beverages: Physicochemical Effects"

_foods, 2020, doi:10.3390/foods9081016_

Round 1

Reviewer 1 Report

Table 1 is confusing and data is repeated in there.

I think it is an interesting research topic. Plant based food beverages are rapidly increasing in the markets and also the benefits of plants are getting more important. The germination of plants leads as the authors stated to beneficial changes in the nutritional values of the plants.
I do not like the referring to the other publications done by the authors, where the methods are described, that might not be open access. It might be my personal opinion, that I do not separate one research topic into several publications. However, the authors know how to write paper and present results, even if I think Table 1 is confusing. I also think that the oil and protein characteristics might be interesting here, as they are also physicochemical.

Author Response

Reviewer #1:

Table 1 is confusing and data is repeated in there.

               Table 1 has a few rows purposely repeated so a few treatments can be compared in different groupings. During the course of several trials with the three rice beverage types, not all data were collected every time, and not all runs could or would be comparable. For example, the beverages prepared from brewers broken white rice (WR) that was not germinated was not run simultaneously and compared to the germinated brown rice (GBR) beverages. Also, different runs oftentimes delivered different flour samples at different times which, were hard to statistically compare. We always used fresh paddy rice to deliver brown rice (BR, which was sometimes run as BRR (brown rice beverages), or germinated into GBR for other beverages. All Rondo rice used was from the same field, same year, field harvest date, same storage conditions, and WR milling by collaborator/supplier, and BRR milling in our pilot plant. We did the best with the data that we could taking into consideration about 35 overall runs were accomplished, and roughly 12 separate trials making three beverage types and “cooked flours” were used to accrue data and get repeats accomplished, through time.

I think it is an interesting research topic. Plant based food beverages are rapidly increasing in the markets and also the benefits of plants are getting more important. The germination of plants leads as the authors stated to beneficial changes in the nutritional values of the plants.

I do not like the referring to the other publications done by the authors, where the methods are described, that might not be open access. It might be my personal opinion, that I do not separate one research topic into several publications.

               The first paper [29.         Beaulieu, J.C.; Reed, S.S.; Obando-Ulloa, J.M.; McClung, A.M. Green processing protocol for germinating and wet milling brown rice beverage formulations: Sprouting, milling and gelatinization effects. Food Sci. Nutr. 2020, 2020, 2445-2457.] is indeed open access. Subsequently, since publication was paid, and open access, we too believe the Supplemental materials herein are readily accessible, and well defined. The first paper had significant focus on the process itself, proximates and some rapid quality appraisal, but not necessarily the physicochemical data and changes through processing. Considering very broad topics (processing protocol development versus changes to macro- and phytonutrients), there was far too much data to submit as one paper. Subsequently, they were originally submitted as two papers, in a series. Our first attempt to publish these was admittedly not strong because I was waiting on potential patent issues through my Agency, which essentially forced me to try hiding much of the methodology. Ultimately, the first article was published, while the second was unsuccessful: due to missing M&M, and needing more robust data (as fixed herein).

However, the authors know how to write paper and present results, even if I think Table 1 is confusing. I also think that the oil and protein characteristics might be interesting here, as they are also physicochemical.

               We have those data however, again, that would present far too much information and be too broad for one paper. Those data are slated to be submitted within 2 months; as concepts expanded and data illustrated how profoundly different this processing protocol is compared to most shelf-stable commercial rice beverages.

Reviewer 2 Report

Overall well-written article. However, the novelty of scientific findings and research gaps in the literature review should be clearly mentioned in the introduction, such as:

1. What are the processing issues during production of rice beverages with GABA?
2. What interventions have been reported in literature to improve the processability of green rice beverages?

Author Response

Reviwer #2:

Overall well-written article. However, the novelty of scientific findings and research gaps in the literature review should be clearly mentioned in the introduction, such as:

  1. What are the processing issues during production of rice beverages with GABA?

               Most articles with germinated brown and colored bran rice indicate that GABA increases. However, it too is known that the efficacy and efficiency varies markedly based on germination conditions (time, temperature, soaking versus germination, per se) and variety etc. Nonetheless. below Table 4 (Lines 470-480), we mentioned a lot of items about GABA, including that GABA generally increases with germination, and germination temperatures above 35 C can decrease GABA (“The maximum germination temperature associated with maximizing GABA concentrations in one short grain japonica and one long-grained indica brown rice was 35 °C, and temperatures above that resulted in GABA decreasing [14].”). We also mentioned (Lines 484-487) that “There is scant information related to tracking some of these nutrients through multi-step processes after germination, such as beverage production that uses markedly higher processing temperatures compared with germination.” Future studies can expand upon temperature effects through pasteurization. Additionally, germinated brown rice has been shown to impart important functional properties in various food applications (Cornejo et al , 2015; Ohtsubo et al., 2005). However, one article indicated that key micronutrient levels were shown to diminish extensively during baking, but control values were not illuminated to see what concentrations existed post-germination (true control) versus after all the baking regimes studied (Cornejo etal 2015). The study of Ohtsubo et al., has commingled methods and different treatment labels that seem convoluted, so much so that a true interpretation of the GABA concentration from control through germination into a single control bread (100% rice) was apparently not made or impossible to decipher. Thus we did not include these references that indeed have treatments with elevated temperatures and GABA analyses.

  1. What interventions have been reported in literature to improve the processability of green rice beverages?

               Line 72. In the 4th paragraph of the Introduction, we let the reader know that there was not a lot of state-of-the-art for comparisons. This too was embellished somewhat for clarity-sake. Thanks for this question!

The literature in this area is mainly patents and a few papers, which mainly focus on enzymatic extractions to keep grain/plant items green and/or solvent-free. The papers generally use hot water and hundred-years old steeping technologies. The patents (e.g. per our self-citation [29. Beaulieu,J.C.; Reed,S.S.; Obando-Ulloa,J.M.; McClung,A.M. Green processing protocol for germinating and wet milling brown rice beverage formulations: Sprouting, milling and gelatinization effects. Food Sci. Nutr. 2020, 2020, 2445-2457.]) did not germinate the brown rice, and either made syrups, white or brown rice beverages, or rice protein isolate through various enzymes. Most patents are purposely difficult to follow, or end up as dead-ends and “patents pending” or proprietary secrets held by companies commercializing the protein isolates. However, we are not aware of any processes whereby non-stabilized brown rice has been minimally treated, processing kept green, true germination occurred (as opposed to only soaking or steeping), had zero additives etc. and delivered a 1000% natural beverage (if one does not consider food-grade enzymes). Subsequently, there appears to be little to no real comparisons.